# Beyond the Crawl: Unmasking Browser Fingerprinting in Real User Interactions

## Abstract

Browser fingerprinting is a pervasive online tracking technique increasingly used for profiling and targeted advertising. Existing research on fingerprinting prevalence relies heavily on automated web crawls, which inherently struggle to replicate the nuances of human-computer interaction. This raises concerns about the accuracy of current understandings of real-world fingerprinting deployments. To that end, this paper presents a user study involving 30 participants over a 10-week period, capturing telemetry data from real browsing sessions across 3,000 top-ranked websites.

Our findings reveal that automated crawls miss nearly half (47.8%) of the fingerprinting websites encountered by real users. This discrepancy mainly stems from crawlers' inability to access authentication-protected pages, circumvent bot detection mechanisms, and trigger fingerprinting scripts activated by specific user interactions. We also identify potential new fingerprinting vectors present in real user data but absent from automated crawls. Finally, we evaluate the effectiveness of federated learning for training browser fingerprinting detection models on real user data, demonstrating superior performance to models trained solely on automated crawl data.

## CCS Concepts

• **Security and privacy** → **Privacy-preserving protocols**; **Privacy protections**; • **Networks** → **Web protocol security**.

## Keywords

Browser Fingerprinting; Differential Privacy; Federated Learning

**ACM Reference Format:**
Anonymous Author(s). 2024. Beyond the Crawl: Unmasking Browser Fingerprinting in Real User Interactions. In *Submission to The Web Conference (TheWebConf'25), April 28–May 2, 2025,Sydney, Australia.* ACM, New York, NY, USA, 12 pages.

## 1 Introduction

Browser fingerprinting is an invasive online tracking technique widely considered a significant threat to users' privacy [9, 26, 55]. Generally, it involves trackers using client-side device information (e.g., hardware specifications or browser configurations) to derive unique device identifiers and track users across multiple visits and websites. Unlike third-party cookies, browser fingerprinting is *stateless* and thus less visible to users, who have limited control and few tools to defend against it.

Worse yet, browser fingerprinting is highly intrusive, as the identifiers remain stable over long periods of time [48] and can

be effective even when using incognito mode [4]. Countermeasures can be tricky to deploy after detection, as they often lead to significant website breakage and affect user experience [17, 31], not to mention that the unique identifier may have already been revealed at that point. With the increasing prevalence of fingerprinting [31] and the phasing out of third-party cookies by major browsers, countering invasive browser fingerprinting has become crucial to protecting user privacy on the modern Web.

Detecting fingerprinting attempts, often implemented through JavaScript scripts, is a prerequisite for effective mitigation. Initially, these scripts were mainly identified through simple heuristics [1, 27] and manually curated blocklists [20, 23, 25]; however, these are hard to maintain and often narrowly defined to reduce false positives, thus missing many fingerprinting scripts in practice. Consequently, machine learning (ML) based detectors have become increasingly popular [18, 29, 31] and can detect significantly more fingerprinting scripts with comparable false positive rates to heuristics. Overall, these techniques generally rely on centralized, automated crawlers instructed to visit a large number of websites to collect scripts for analysis and fingerprinting detection.

One major limitation of automated crawlers is their limited ability to faithfully replicate genuine user behaviors and interactions. Although recent work has made progress toward emulating some degree of user interactions (such as accepting cookie banners [34, 51] and emulating user interests in browsing patterns [34]), many key actions remain firmly outside of the reach of automated crawlers. For instance, these often fail to reliably solve CAPTCHAs, evade bot detectors, login, access websites behind paywalls, etc. This leads to incomplete website coverage and potentially missed fingerprinting scripts. For example, Iqbal et al. [31]'s automated crawl failed to visit 11.9% of top-ranked websites.

To address this limitation, prior work has investigated the feasibility of training ML models on real-user browsing sessions. Evidently, this needs to be done in a privacy-preserving way to avoid exposing users' browsing histories, etc. Annamalai et al. [8] recently proposed FP-Fed, a system relying on differentially private federated learning (DP-FL) to collaboratively train ML models on the combined browsing sessions of many users while providing strong formal privacy guarantees. Although FP-Fed can train models that achieve good performance even at moderate levels of privacy, the authors only tested it on a dataset derived from an automated crawl, with unclear implications for its performance in the real world.

**Technical Roadmap.** In this paper, we investigate the prevalence and distribution of browser fingerprinting in real-user browsing sessions, as opposed to automated crawls used in prior work [8, 27, 31]. Specifically, we collect real-world data from a 10-week study (June-August 2024) involving 30 participants who browsed the Web with their own devices and reported telemetry via a Chrome extension. This enabled us to collect browser fingerprinting signals from real browsing sessions across 3,000 top-ranked websites.

This study allows us to shed light on two distinct aspects of fingerprinting. First, we compare the prevalence and distribution of browser fingerprinting in a real-world dataset to an equivalent automated crawl of the same 3,000 websites. Second, we conduct a comparative analysis of the ML performance of browser fingerprinting detectors trained on the automated crawl and the real-world dataset in a distributed and privacy-preserving way.

**Main Findings.** Overall, our study and experimental analysis yield three interesting findings:

- Surprisingly, *almost half* (47.8%) of fingerprinting websites identified from real-user browsing sessions are missed by automated crawls. This is due to three main reasons: (1) authentication pages that are typically not crawled by automated crawlers, (2) bot detection scripts that track real user interaction before browser fingerprinting is triggered, and (3) cookie banners that require user consent before performing fingerprinting.
- We discover potential new fingerprinting vectors from the real-user browsing sessions that were not previously found in the automated crawls.
- We show that ML models trained privately on real-user browsing sessions can achieve comparable or even better performance than non-private models trained on automated crawls alone while providing privacy and learning the behaviors of many more fingerprinting scripts. Specifically, in our experiments, the former achieves an Area Under the Precision-Recall Curve (AUPRC) of 0.982 at a privacy level of $\varepsilon = 5$, compared to an AUPRC of 0.971 for the latter.

Overall, our work paves the way for large-scale deployment of more effective, dynamic, and robust browser fingerprinting detection relying on a scalable, distributed, and privacy-preserving infrastructure geared to be readily integrated into modern browsers.

## 2 Background & Related Work

In this section, we review browser fingerprinting and differentially private federated learning, along with relevant prior work.

### 2.1 Browser Fingerprinting

Browser fingerprinting is a tracking technique usually deployed through Javascript running on a user's browser to build a unique user identifier. This user identifier typically consists of high-entropy device information (e.g., screen size, GPU model) used to produce highly unique and stable user identifiers. Unlike third-party cookies, browser fingerprinting is *stateless*, i.e., no data is stored on the user's device and, therefore, cannot be easily detected or mitigated. Although browser fingerprinting has been known to be used for legitimate purposes, e.g., Web authentication [5, 33, 51] or fraud detection [30, 35], it is often used for online tracking and to serve targeted ads [34]. As a result, browser fingerprinting is widely considered a significant threat to user privacy [55], thus prompting many browser vendors to deploy countermeasures [9, 12, 26].

Although there are well-documented cases of fingerprinting in the wild, we are not aware of a widely accepted formal definition of fingerprinting [31]. Mayer [36] was the first to observe that the uniqueness and customization of browsing environments (which they called "quirkiness") can be abused to identify users. The large-scale Panopticlick experiment later conducted by the late Peter

Eckersley [24] showed that most browsing sessions (83.6%) have unique fingerprints. While these results are concerning on their own, the fingerprinting surface considered by this early work was relatively limited to information collected from simple Javascript APIs like the `Screen` and `Navigator` APIs and HTTP headers. More recently, as more features and APIs were added to the Javascript specification, fingerprinting has begun to include Canvas [40], Audio [27], WebRTC [27], WebGL [13], Battery Status [45], Mobile Sensor [11], and even the Web Bluetooth APIs [10]. With such a large fingerprinting surface, it is often difficult to pinpoint the *intent* behind the use of these various APIs in arbitrary Javascript scripts, thus making it challenging to specify a single comprehensive definition of fingerprinting.

Nevertheless, we follow prior work by Iqbal et al. [31] and take a conservative approach to defining fingerprinting based on a well-known high precision (low false positive rate) heuristic. Specifically, we exclude the simple curation of properties from the `Navigator` and `Screen` APIs and focus on the four most obvious forms of fingerprinting (Canvas, Canvas Font, Audio, and WebRTC). This definition not only minimizes the probability of flagging non-fingerprinting scripts but is also helpful in training ML models that generalize well to other forms of fingerprinting as well [8].

Next, we briefly describe the four main types of fingerprinting identified by the heuristics [27] and their associated criteria.

***Canvas.*** Scripts exploit differences in how fonts are rendered across devices. Criteria:
1. text is written to the `canvas` element using the `fillText` or `strokeText` method;
2. style is applied with `fillStyle` or `strokeStyle` method;
3. `toDataURL` is called to extract the image from the `canvas`; and
4. `save`, `restore` or `addEventListener` methods are not called.

***Canvas Font.*** Scripts access the list of fonts installed on a device. Criteria:
1. `font` property of canvas element is set to more than 20 different fonts; and
2. `measureText` is called more than 20 times.

***WebRTC.*** Scripts rely on the uniqueness of peers in the WebRTC protocol [41]. Criteria:
1. `createDataChannel` or `createOffer` method is called on a WebRTC peer connection; and
2. `onicecandidate` or `localDescription` method is called.

***AudioContext.*** Scripts exploit differences in the different hardware processing audio. Criteria:
1. either `createOscillator`, `createDynamicsCompressor`, `destination`, `startRendering`, or `oncomplete` are called.

### 2.2 Detecting Browser Fingerprinting

Early research on browser fingerprinting detection has mainly relied on manually curated blocklists, e.g., EasyPrivacy [23], Privacy Badger [25], and Disconnect [20]. However, as fingerprinting vendors and scripts constantly evolve, it can be challenging to maintain these lists continuously.

**Heuristics-based detection.** Manual analysis of fingerprinting scripts by Acar et al. [2] and Roesner et al. [49] paved the way

for the first heuristic that detected canvas fingerprinting automatically [1]. More precisely, Acar et al. [2] monitored and analyzed the arguments and return values of the `fillText`, `strokeText`, and `toDataURL` methods exposed by the Canvas API. Three additional types of fingerprinting (Canvas Font, WebRTC, and Audio) were later added by Englehardt and Narayanan [27]. This combined heuristic is now widely used as a prominent indicator of fingerprinting [8, 18, 31, 51] as it is known to produce very few false positives, which is an important consideration to prevent falsely flagging fingerprinting scripts. However, as noted by Iqbal et al. [31], it might miss many fingerprinting scripts in practice since it is defined very narrowly to achieve high precision. Furthermore, keeping the heuristic up-to-date with the latest Javascript APIs and fingerprinting vectors can be challenging.

**ML-based detection.** Methods based on machine learning solve the problem of manually maintaining blocklists and heuristics by learning fingerprinting behaviors in the wild. Ikram et al. [29] trained a one-class SVM on *static* features directly extracted from scripts' source code. However, Iqbal et al. [31] observed that code is often obfuscated, making it difficult to reliably learn fingerprinting behaviors from static features alone; therefore, they additionally trained a Decision Tree classifier on *dynamic* features extracted from the execution trace of a given script. By monitoring and analyzing the number of times a script calls each Javascript API, along with the associated arguments and return values, Iqbal et al. [31] trained a robust browser fingerprinting detector that achieves both high precision and high recall.

Recently, Annamalai et al. [8] proposed going beyond *centralized* models that rely on a large automated crawl, which cannot replicate human interaction and, therefore, might miss fingerprinting scripts in the wild. They proposed FPFed, a system geared to train a browser fingerprint detection model collaboratively on real users' browsing sessions while preserving privacy using the federated learning paradigm [37] (see below). However, Annamalai et al. [8] did not actually evaluate their system on real-world user behavior but opted to test their system by simulating real-world users on an automated crawl. Our work aims to fill this research gap by collecting browsing data from real users and evaluating the FP-Fed system on a real-world dataset instead.

## 2.3 Differentially Private Federated Learning

The standard way to train models collaboratively on multiple users' data while providing formal privacy guarantees is through differentially private federated learning (DP-FL). More precisely, DP-FL combines Federated Learning (FL) and Differential Privacy (DP).

**Federated Learning.** Typically, centralized ML models are trained on datasets stored at a single entity. However, in many scenarios, gathering data from multiple users may not be possible due to privacy, security, and/or efficiency concerns. To this end, Federated Learning (FL) [37] introduces a distributed learning approach allowing users to train an ML model collaboratively without disclosing their (potentially sensitive) training data.

In FL, users train local models on their individual training data, sharing only their model updates (and not the raw training data) with a server. The server aggregates the model updates to build a global model, which is then propagated back to the users. This process then repeats until the global model converges. Although FL ensures that the server never sees the raw training data, prior work has shown that the model updates can still leak sensitive information about users' training data [39]. Therefore, recent work has focused on providing formal privacy guarantees.

**Differential Privacy (DP).** DP is the standard framework for defining algorithms that provide theoretical upper bounds on the loss of privacy incurred by data subjects due to the output of an algorithm [22].

*Definition 2.1 (Differential Privacy (DP)).* A randomized mechanism $\mathcal{M} : \mathcal{D} \rightarrow \mathcal{R}$ is $(\varepsilon, \delta)$-differentially private if for any two neighboring datasets $D, D' \in \mathcal{D}$ and $S \subseteq \mathcal{R}$

$$\Pr[\mathcal{M}(D) \in S] \leq e^\varepsilon \Pr[\mathcal{M}(D') \in S] + \delta$$

The privacy guarantees provided by a differentially private algorithm are characterized by the parameters $\varepsilon$ and $\delta$. The privacy parameter, $\varepsilon$, is often referred to as the privacy budget and ranges from 0 to $\infty$, with lower values denoting better privacy. Whereas $\delta$ quantifies the probability that the mechanism fails to deliver any guarantees; typically, $\delta$ is fixed to some asymptotically small number (e.g., $10^{-5}$).

**DP-FL.** DP can be combined with FL in many ways to provide strong, formal privacy guarantees in the FL setting. One way is by adding statistical noise when the server aggregates the model updates, aka Central DP (CDP). Under this regime, the model is trusted with the aggregated model updates but not the raw sensitive data. CDP guarantees that it is impossible (up to the privacy parameter $\varepsilon$) to infer whether or not data from a user was used to train the global model based on the aggregated and noised model updates. Examples of CDP instantiations for DP-FL include next-word prediction [38], medical image analysis [3], and network analysis [42]. Other approaches use Local DP [52, 54] or Distributed DP [32] to minimize trust assumptions in the server.

In this work, we consider the FP-Fed [8] setting, which uses CDP, as it provides a good trade-off between utility, convergence speed, and the amount of noise required for the desired level of privacy. (A detailed overview of the FP-Fed system can be found in Appendix A.)

## 3 Methodology

Next, we discuss our methodology to collect browser fingerprinting scripts from real users' browsing sessions. We describe the websites we collect data from and how we recruit participants, collect website telemetry, and detect fingerprinting. We also discuss relevant ethical considerations with respect to collecting and analyzing the data.

## 3.1 Websites of Interest

Ideally, to capture real user interactions, we would want to collect and analyze telemetry data from all websites visited by participants. However, browsing histories are highly sensitive, and collecting them might violate participants' privacy, discourage them from contributing to the study, or even introduce biases in the data collection. (Note that while FL-based detection would ensure that raw telemetries are not disclosed, we do need them for our experiments when measuring accuracy and finding discrepancies).

As a result, we opt to collect telemetry from a selected set of websites–more specifically, the top-ranked websites according to the Chrome User Experience Report (CrUX) from April 2024 [14]. Overall, we collect data from 3K websites sampled from the top 5K ranked websites in the CrUX report: specifically, we take the top 1K ranked websites and a random 2K sample of top 1K to 5K ranked websites. To further limit the amount of sensitive information collected, we use Cloudflare's Domain Intelligence API [15] to detect and filter out websites containing adult or potentially harmful content. Specifically, we exclude websites from the following categories (as defined by the Domain Intelligence API): 1) *Adult Themes*, 2) *Gambling*, 3) *Questionable Content*, 4) *Security Threats*, 5) *Violence*, and 6) *Security Risks*. Out of the top 5K ranked websites from the CrUX ranking, we excluded 948 such websites, before sampling the 3K websites that we use in our study.

## 3.2 Participant Recruitment

We recruited 30 participants (aged 18 to 60) who used the Chrome browser through the Amazon Mechanical Turk (MTurk) platform [6]. The "Human Intelligence Task" (HIT), i.e., the ad used to recruit users on MTurk, can be found in Appendix B.

Each participant was instructed to download a Chrome extension we developed and given a password to authenticate on the extension. As mentioned above, for privacy purposes, each participant was provided with a pre-defined set of 100 websites to visit instead of collecting data from their "natural" browsing sessions. (Note that the extension only collects data from this list.)

For each website, the participants were instructed to visit at least ten sub-pages (including login pages), accept cookie banners, and solve CAPTCHAs when presented to simulate realistic browsing sessions. The extension collected the telemetry from the websites in the background and transmitted it to our server. Other than the website telemetry required for the purpose of fingerprinting detection, no other data (e.g., demographics, name, email) was collected. Upon successfully visiting all of the given websites, the participants were given a "Task Completion Code" by the extension, which was then used to compensate them through MTurk. In total, the participants received an average of USD 20.33 (excluding taxes and fees), which exceeds the federal minimum wage.

Although only a relatively small number of participants were recruited, we emphasize that the focus of this study is not to draw conclusions about the distribution or popularity of websites as visited by real users in their natural browsing behaviors. Specifically, the intent of this study is to investigate to what extent automated crawlers miss fingerprinting scripts due to their inability to replicate real human interactions in practice. To that end, this study could have been equivalently performed by a single participant crawling all 3K websites, but this would have made it hard to recruit a participant to complete the task successfully.

**Data Quality.** Overall, the participants visited a total of 14,895 unique URLs. In Figure 1, we plot the distribution of unique URLs visited by each participant. While the extension verified that each participant visited all of the websites assigned to them before providing the "Task Completion Code," we did not verify if each participant did visit at least ten sub-pages on each website as this would have been difficult to do so in general (e.g., single page applications

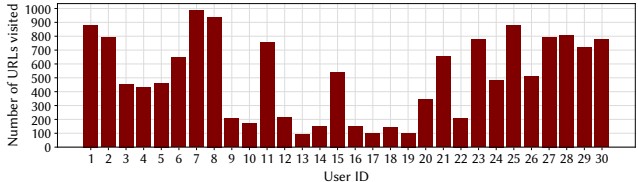

**Figure 1: Distribution of unique URLs visited by each study participant.**

may not redirect to different URLs). On average, each participant visited 506 unique URLs. This is five times the number of unique URLs visited by automated crawlers in prior work [27, 31].

## 3.3 Collecting Scripts and Extracting Features

Key to identifying browser fingerprinting is dynamically analyzing scripts that are loaded by the websites [31, 43]. To collect this data, as mentioned, we built a Chrome extension that monitors and records the Javascript APIs accessed by each script loaded on a website, along with the associated arguments and processed return values. Our extension then processes the collected data and sends telemetry to our server, where we then analyze it and detect fingerprinting scripts. The extension only collects data from the domains corresponding to the list of websites assigned to each participant, i.e., it does not gather data from other websites, which the participant might visit in other browsing sessions.

We built our extension based on the instrumentation developed by Iqbal et al. [31] for automated crawls with the Mozilla Firefox browser. We adapted the instrumentation to work with the Google Chrome browser instead, extending it to monitor Chrome-specific APIs as well, and injected the instrumentation using a Chrome extension that participants can easily install.

Additionally, to reduce storage and data transmission costs, our extension first pre-processes the raw data and extracts only the features necessary to detect fingerprinting. These features consist of API call counts (i.e., how many times each Javascript API is called) and "custom" features processed from the arguments and return values (e.g., length of string argument, number of elements in returned list value). This pre-processing step not only reduces the data transmission costs but also ensures that the collected data is more private, as the exact arguments and return values are not sent to the server. In summary, our extension processes the websites visited by the participant, the scripts loaded by each website, extracts features from each script's execution trace, and sends this data to our server for analysis.

## 3.4 Fingerprinting Detection

We follow prior work on browser fingerprinting detection [8, 27, 31] and use high-precision heuristics to label scripts as *fingerprinting*. Specifically, we use the heuristic developed by [27] and later modified by [31]. We follow this conservative approach to labeling fingerprinting scripts over machine learning classifiers [31] to ensure low false positive rates and have good confidence that the labeled scripts are, in fact, fingerprinting. The heuristics we use identify four main types of fingerprinting – namely, Canvas, Canvas Font, WebRTC, and Audio Context (see Section 2.1 – and does not

consider simple accesses to device information as fingerprinting to minimize false positives.

## 3.5 Ethics

Our study was reviewed and approved by our Institutional Review Board (details are omitted to preserve submission anonymity). As part of the process, we submitted the full documentation detailing our approach to recruiting and compensating participants, the types of data that will be collected, along with data access policies, potential ethical issues, mitigation strategies, as well as the information sheet and consent form provided to participants.

All participants were recruited and compensated entirely anonymously through the Amazon MTurk platform. Specifically, each participant received an average of USD 20.33 and was conservatively expected to take at most an hour to complete visiting all 100 websites assigned to them. Therefore, the compensation was well above minimum wages both in California [44] (USD 16/hour) and the United Kingdom [16] (GBP 11.44/hour).

In line with data minimization principles, we only collected pre-processed and extracted features from the scripts' execution traces. Specifically, we did not collect the raw return values and arguments of the Javascript APIs and did not collect any other user data or metadata, e.g., IP address, device, or network information.

As mentioned, we provided participants with a list of websites to visit instead of monitoring their "natural" browsing sessions, and designed our Chrome extension to only collect data from this list. That is, on all other websites, the extension does not inject the instrumentation script and does not monitor or record any information from these websites. By doing so, we can filter out potentially embarrassing or dangerous websites (e.g., adult entertainment, spam) and prevent accidentally collecting data from participants' visits to such websites.

Additionally, the exact types of data collected along with the purpose of the data collection, potential privacy implications of taking part in the study, and participants' data rights (e.g., withdrawal from data collection) were explained in layman terms through a participant information sheet which they could download. Furthermore, explicit user consent was obtained through a consent form, which reiterated their rights and privacy implications of taking part. Overall, we abided by a strict code of ethics (i.e., RESPECT [28]). In line with our institution's data retention policies, we will delete all data within ten years of the publication of our results.

## 4 Results

In this section, we present the results of our study. We begin by comparing the prevalence of browser fingerprinting found in real user browsing sessions with that observed in an automated crawl of the same 3K top-ranked websites. Next, we shed light on why some fingerprinting websites are missed by automated crawlers, focusing on user interactions and website categories. Finally, we compare the prevalence of Javascript APIs in fingerprinting scripts collected from real user browsing sessions with that of the automated crawl and discover potential new fingerprinting vectors.

|  | Scripts (FP / Total) | Websites (FP / Total) |
|---|---|---|
| Real Users | 695 / 80,969 | 471 / 3,000 |
| Automated Crawl | 210 / 32,823 | 363 / 3,000 |

Table 1: Number of scripts and websites detected as fingerprinting by real user browsing session vs automated crawl

### 4.1 Prevalence of Fingerprinting

To compare against an automated crawl, we follow the same strategy as Annamalai et al. [8], using an instrumented Chrome browser along with Puppeteer to visit the same 3K top-ranked websites provided to the participants of our study. We also try to simulate some degree of user interaction (i.e., scrolling and taking a full-page screenshot) in the automated crawl, but the crawler does not visit any subpage linked from the main website.

In Table 1, we report the number of fingerprinting scripts collected during the study with real users vs. the automated crawl, along with the number of websites that load these fingerprinting scripts. We also defer a more comprehensive breakdown of the fingerprinting scripts identified to Appendix C.

First, we detect 2.3x more fingerprinting scripts through real-user browsing sessions than through the automated crawl. This suggests the latter may indeed miss many fingerprinting scripts in practice, corroborating preliminary findings by Annamalai et al. [8], who show similar results from a small set of websites. However, this increase may be due to the overall increase in the number of scripts collected from real-user browsing sessions on the same set of websites; specifically, we observe a 146% increase in the number of scripts (80.9k vs. 32.8k) compared to a 34% increase in the number of scripts that are fingerprinting (i.e., 0.86% vs. 0.64%). Nevertheless, we observe a significant increase in the percentage of fingerprinting *websites*; specifically, out of the same 3K websites, we identify 108 more fingerprinting websites (a 30% increase) from real-user browsing sessions than from automated crawlers.

We also find that real-user browsing sessions missed 117 fingerprinting websites that were instead detected from automated crawls. However, this was due to a bug in the implementation of our Chrome extension that prevented the instrumentation script from being injected into iframes, which are often used for browser fingerprinting. In other words, this was an isolated bug and not an inherent limitation of distributing the deployment of browser fingerprinting detection to end users.

Taking the bug into account, the automated crawl missed 225 fingerprinting websites that were identified from real user browsing sessions. This confirms that automated crawlers cannot replicate real user interactions, which can result in the prevalence of fingerprinting being potentially underestimated in practice.

### 4.2 Undetected Fingerprinting on Websites (by Sub-Pages)

Next, we take a closer look at the user interactions that specifically trigger fingerprinting scripts on websites. To that end, we group the 225 fingerprinting websites undetected by the automated crawler

| Failed | Auth | Content | Home | **Total** |
|--------|------|---------|------|-----------|
| 30 | 15 | 63 | 117 | **225** |

**Table 2: Number of fingerprinting websites undetected with automated crawl broken down by reason.**

by the specific sub-pages that loaded fingerprinting scripts and report this in Table 2.

**Failed Visits.** First, we find that the automated crawler failed to visit 30 out of the 3K websites (1%). We believe this is due to bot detectors running *before* the page is loaded; in fact, the majority of these websites (66.7%) return 4XX errors that can be associated with bot detection scripts [8].

**Authentication & Content Pages.** We find that a non-negligible number of fingerprinting scripts appear on authentication pages (e.g., login, account, and sign-up pages), and the automated crawler only visited 17 such pages as opposed to the 240 visited by real users. As a result, the automated crawler missed 15 websites that were deploying fingerprinting scripts on the authentication page.

Similarly, as the automated crawler did not visit any inner content pages, it missed another 63 websites that were fingerprinting on the content page. While recent work [51] shows that it is possible for automated crawlers to infer whether a visited page is an authentication page using ML-based methods, we note that such pages might themselves deploy bot detection scripts preventing automated crawlers from successfully visiting them. In fact, our automated crawler failed to visit 6 such authentication pages (26.1%).

**Home Pages.** The automated crawl also missed fingerprinting scripts on 117 home pages. This is surprising as these scripts were not triggered even though the automated crawler successfully visited these pages. This indicates that in some cases, even if the website is loaded successfully, specific user interactions are required to trigger fingerprinting scripts. To investigate this more deeply, we manually visited 11 of these websites (a 10% sample) to observe the specific user interactions that triggered the fingerprinting scripts. We found that 5 out of the 11 websites deployed bot detection scripts *after* the websites were loaded (as opposed to *before*, which we previously categorized as a "failed visit"). Popular vendors for this kind of script include PerimeterX[1] (now called Human) and SiftScience[2]. Therefore, the fingerprinting scripts were no longer triggered because the automated crawlers were detected as bots. Also note that four websites only start fingerprinting after user consent is received from a cookie consent banner, consistent with findings from prior work [46]. For the remaining two pages, we were unable to pinpoint the exact user interaction that triggered browser fingerprinting, as we could not consistently get the fingerprinting script to load.

In summary, even if automated crawlers could crawl the Web more deeply and mimic certain user interactions, they are often caught out by advanced bot detectors and may not be able to trigger fingerprinting scripts that require specific user interactions.

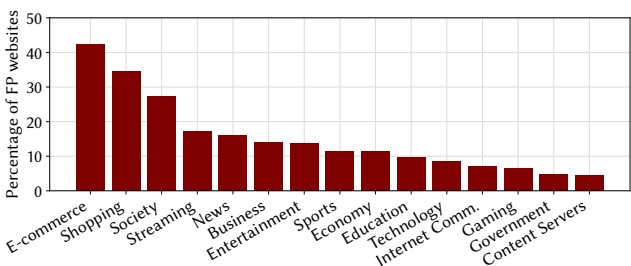

**Figure 2: Percentage of fingerprinting websites for each website category.**

## 4.3 Undetected Fingerprinting on Websites (by Category)

Next, we analyze the prevalence of fingerprinting websites by category[3] in real-user browsing sessions, as presented in Figure 2. We also quantify the prevalence of fingerprinting by website category in the automated crawl; however, due to space limitations, we omit the results here, as we observe no significant difference between the general prevalence of fingerprinting in real-user browsing sessions and the automated crawl. As opposed to prior work [27, 31], we find that E-Commerce, Shopping, and Society (& Lifestyle) categories have much higher rates of fingerprinting than News websites. We believe one of the main reasons for this discrepancy is that previous studies used the discontinued Amazon Alexa ranking [7], whereas we use the CrUX ranking, which is maintained and known to reflect real-user browsing patterns more accurately [50]. Additionally, as no significant difference was observed in the prevalence of fingerprinting by website category between our automated crawl and the real-user browsing sessions, we do not believe this discrepancy was due to real user behaviors. Note that other categories (e.g., adult content) are also not represented in our results, as we had filtered out these websites for privacy reasons.

Finally, we observe that fingerprinting is more likely to be undetected by the automated crawler for a few specific categories of websites than others. To measure this, we introduce the notion of *Miss Percentage*, defined as the percentage of websites detected as fingerprinting by the real-user browsing sessions but undetected by the automated crawl. Figure 3 shows that Technology and Video Streaming fingerprinting websites are the most likely to be missed by automated crawlers than by real-user browsing sessions. This is expected as Technology and Video Streaming websites might require user interaction or user login before entering the "main page" of the application, where fingerprinting is expected to happen.

## 4.4 Comparison of Fingerprinting APIs

Next, we investigate the differences in the Javascript APIs used frequently by the fingerprinting scripts captured from real-user browsing sessions, as compared to those captured from automated crawls. To do so, we quantify the relative prevalence of APIs used for fingerprinting following prior work by Iqbal et al. [31] and compute the *Call Ratio* for each API found in the real-user browsing sessions, i.e., the ratio between the number of times a given API is called by

---

[1]https://www.humansecurity.com/
[2]https://sift.com/

[3]We categorize the websites using the Cloudflare Domain Intelligence API [15].

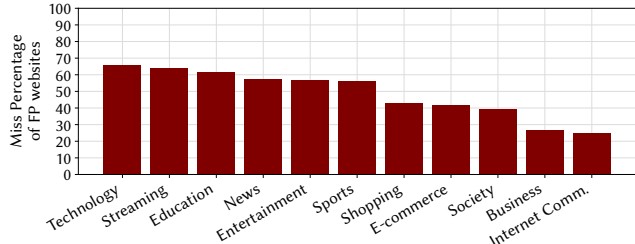

**Figure 3: Percentage of fingerprinting websites undetected by the automated crawler in each website category.**

| Javascript API | Call Ratio |
|---|---|
| audiocontext.sinkid | ∞ |
| audiocontext.onsinkchange | ∞ |
| rtcpeerconnection.getconfiguration | ∞ |
| rtcpeerconnection.sctp | ∞ |
| svgtextcontentelement.getextentofchar | ∞ |
| rtcpeerconnection.tostring | ∞ |
| window.navigator.plugins[chrome pdf plugin] | 7.50 |
| window.navigator.plugins[webkit built-in pdf] | 6.77 |
| window.navigator.plugins[microsoft edge pdf viewer] | 6.77 |
| window.navigator.plugins[chrome pdf viewer] | 6.77 |
| window.navigator.plugins[chromium pdf viewer] | 6.77 |
| window.navigator.plugins[pdf viewer] | 5.65 |
| offlineaudiocontext.hasownproperty | 3.62 |

**Table 3: Call Ratio of a sample of Javascript APIs predominantly used by fingerprinting scripts in real-world browsing sessions.**

fingerprinting scripts and by non-fingerprinting scripts. The Call Ratio is high (above 1) when a given API keyword is used more prevalently in fingerprinting than non-fingerprinting scripts.

Specifically, we use it to identify potential fingerprinting vectors from real-user browsing sessions that automated crawlers might otherwise miss. In Table 3, we report the APIs with high Call Ratios in the real-user browsing sessions that were simultaneously not used by any fingerprinting scripts captured in the automated crawl (i.e., the API would have been missed by the automated crawl). The call Ratio is ∞ when no non-fingerprinting scripts use the keyword.

We observe that audio and WebRTC fingerprinting are more prevalent in real-user browsing sessions than automated crawls. Specifically, we observe that multiple Audio APIs (i.e., audiocontext.sinkid, audiocontext.onsinkchange) and WebRTC APIs (i.e., rtcpeerconnection.getconfiguration, rtcpeerconnection.sctp, rtcpeerconnection.tostring) are exclusively used by fingerprinting scripts in real-user browsing sessions (Call Ratio = ∞). At the same time, these APIs did not appear to be used by any fingerprinting script in the automated crawl. This is probably due to these fingerprinting techniques only occurring if audio devices or peers are present in the network and triggered, which requires real-user devices with an audio interface or a crawler setup that simulates them effectively. Nevertheless, prior work [27] has identified the AudioContext and RTCPeerConnection APIs as prominent fingerprinting vectors.

On the other hand, accesses to the Navigator API has previously not been considered a robust signal to detect fingerprinting [27, 31] as they are often used by non-fingerprinting scripts as well. Conversely, our analysis of real-user browsing sessions shows that specific attributes of the Navigator API can, in fact, be used as reliable signals. Specifically, from Table 3, we note that the PDF viewer plugin is used predominantly by fingerprinting scripts to identify the specific browser being used. Upon closer inspection, we observe that fingerprinting scripts use the "length" and "description" attributes of these plugins to identify which PDF viewer is being used by the browser. Although this API is now deprecated, it is still available in all modern browsers [21], and we believe that the differences in how the specification is implemented across different versions of browsers (e.g., returning a hard-coded list) might be a useful fingerprinting vector.

Overall, this confirms that real-user browsing sessions shed better light on fingerprinting vectors used in the wild compared to automated crawlers.

## 5 Privacy-Preserving Federated Browser Fingerprinting Detection

In the previous section, we showed that automated crawls might indeed miss fingerprinting scripts that are instead captured by real-user browsing sessions. This sheds light on the benefits of training fingerprinting detection models on telemetry from real-user browsing sessions. However, having users share this data with a third party would be both inefficient and extremely privacy-invasive. On the other hand, only training locally (i.e., each user trains their own local model) would yield low accuracy and result in losing crucial knowledge about new fingerprinting behavior.

Recently, Annamalai et al. [8] introduced FP-Fed, a system allowing participants to collaboratively build a browser fingerprinting detection model while keeping their data private, but only sharing model updates using the Federated Learning paradigm [37]. However, they ultimately only tested it on an automated crawl (i.e., they did not capture real user interactions). In the rest of this section, we investigate whether FP-Fed is effective at training browser fingerprinting detectors on real-user datasets.

### 5.1 Overview of FP-Fed [8]

In a nutshell, FP-Fed works as follows. First, participants train local models to detect browser fingerprinting based on the data collected from their individual browsing sessions. The participants then share the model updates (but not the raw collected data) with a central server, which aggregates the model updates and adds noise to satisfy differential privacy and protect participants' privacy. This differentially private "global" model is then shared with the participants, and the process repeats over multiple rounds until the global model converges. Please see Appendix A for more details.

**Improvements.** In our work, we do not only deploy FP-Fed but also modify it to improve model performance. Specifically, instead of training the local models from scratch, as done in [8], we first pre-train the local models, non-privately, on public data and then fine-tune them on the private local training datasets. This is a popular approach used to privately train ML models that yields significantly better model performance [19, 53].

| FP-Fed on Real-User Sessions | | | Automated |
|---|---|---|---|
| $\varepsilon = 1$ | $\varepsilon = 5$ | $\varepsilon = 10$ | (Centralized) |
| $0.979 \pm 0.013$ | $0.982 \pm 0.013$ | $0.983 \pm 0.011$ | $0.971 \pm 0.021$ |

**Table 4: AUPRC of Fed-FP [8] at various levels of privacy ($\varepsilon$) on real-user browsing sessions vs. a centralized model built on the automated crawl.**

In our experimental evaluation, we first pre-train the local models on an equivalent automated crawl of the 3K top-ranked websites. As this is public data, pre-training local models on this data can be done non-privately without degrading the privacy guarantees provided by differential privacy. Furthermore, by combining the automated crawl and real-user browser sessions, we argue that the model can achieve precision that is close to models built only on the automated crawl from prior work [31], while learning from many more fingerprinting scripts present in the real-user browsing sessions thus improving the model's recall. We test this claim empirically next.

### 5.2 Evaluation

We now analyze the feasibility and effectiveness of training detection models for browser fingerprinting using FP-Fed [8]. To do so, we first partition the scripts from real-user browsing sessions into 80% for training (64,776 scripts, 556 fingerprinting) and 20% for testing (16,193 scripts, 139 fingerprinting). Next, we re-sample the training data using the fine-grained Tranco [47] ranking as done in [8] to simulate 1 million users participating in FP-Fed, as expected in a real-world scenario. Finally, we train a Logistic Regression classifier using FP-Fed at various privacy levels and test the model performance on the test set. As mentioned above, recall that we first pre-train the local models of participants with the scripts collected from the automated crawl.

In Table 4, we report the Area Under Precision-Recall Curve (AUPRC) statistic along with the standard deviation over 5-fold stratified cross-validation to assess model performance. We choose AUPRC as it summarizes the model performance over many possible thresholds that can be tuned to achieve a desired precision/recall trade-off. Furthermore, to assess the robustness of the model in a potential real-world deployment scenario, we compare its performance in two distinct cases. In the first one, we start by training a base model (centrally, no federation) with data from the automated crawl and then fine-tune it with FP-Fed on actual data from real-user browsing sessions. In the second case, which corresponds to the traditional approach commonly used in the literature, we train the central model with data from the automated crawl alone. To provide a realistic performance assessment on real-world data, we evaluate both models on the 20% testing set of the real-user browsing sessions.

Our results show that even in high privacy settings ($\varepsilon = 1.0$), the model trained on real-user browsing sessions reaches an AUPRC = 0.979, with performance slightly exceeding the centralized model trained with no privacy on the automated crawl only, which achieves AUPRC = 0.971. This is notable considering that the noise introduced by DP makes it challenging for private classifiers to achieve

even comparable performance to their non-private counterparts [19]. In fact, even at a moderate privacy level ($\varepsilon = 5.0$), the model trained on real-user browsing sessions outperforms the automated crawl by more than one percentage point.

This confirms that the federated model can leverage the fingerprinting behaviors of the scripts present in the combined browsing sessions from real users that are otherwise unavailable to models trained on the automated crawls alone. While this result is promising, we note that the standard deviations we observe are relatively large, which calls for further research covering a larger number of websites and users.

### 6 Conclusion

This paper presented the design and execution of a user study geared to investigate the differences in the prevalence and distribution of browser fingerprinting in real-user browsing sessions as opposed to automated crawlers predominantly used by prior work in browser fingerprinting [27, 31]. To do so, we built a Chrome extension and collected fingerprinting scripts from 30 participants as they browsed 3,000 top-ranked websites. We compared the resulting differences by simultaneously performing an automated crawl of the same websites. Additionally, we evaluated the feasibility and effectiveness of collaboratively and privately training a distributed browser fingerprinting detection model using federated learning.

Our analysis showed not only that automated crawls missed a non-negligible amount of fingerprinting scripts but that they also heavily underestimated the prevalence of browser fingerprinting in top-ranked websites. Specifically, we observed that 47.8% of websites identified as fingerprinting from real-user browsing sessions were undetected by the automated crawl. Our findings empirically validate existing hypotheses that discrepancies arise because automated crawlers lack the behavioral nuances of human users. Consequently, bot detection scripts often block them or fail to trigger website fingerprinting mechanisms. Finally, we showed that ML models trained on a combination of crawled data and subsequently fine-tuned in a privacy-preserving way with data from real browsing sessions on-device likely yield better performance on real-world datasets than models trained purely on crawled data.

**Limitations.** Naturally, our work is not without limitations. For instance, the scope of website coverage in our study is limited. While our analysis encompasses an order of magnitude more websites than previous research [8], it represents a subset of the websites visited daily by users. This limited scope was deliberate and focused on protecting the privacy of our participants, as ultimately, the study's primary objective was to investigate discrepancies in fingerprinting prevalence and distribution between real-user browsing sessions and automated crawls.

Having established a demonstrable difference, future research should prioritize expanding the study's reach by increasing both the number of websites analyzed and the participant pool. Crucially, this expansion will necessitate further development of privacy-preserving data collection and analysis techniques, a direction we intend to explore in future work.

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

## A  FP-Fed [8]

FP-Fed [8] is a privacy-preserving distributed system for detecting browser fingerprinting in the wild. Using Differentially Private Federated Learning (DP-FL) [37], FP-Fed enables learning the fingerprinting behaviors of scripts from the combined browsing sessions of many users while providing strong formal guarantees of privacy. There are five main steps to FP-Fed, also depicted in Figure 4:

(1) Participants run an instrumented browser (e.g., with a Chrome extension installed) and naturally browse websites.
(2) The instrumented browser / extension collects script execution data and performs the following actions before federated training begins:
    (a) Collects number of times specific (often high-entropy) monitored APIs are called, along with associated return values and arguments;

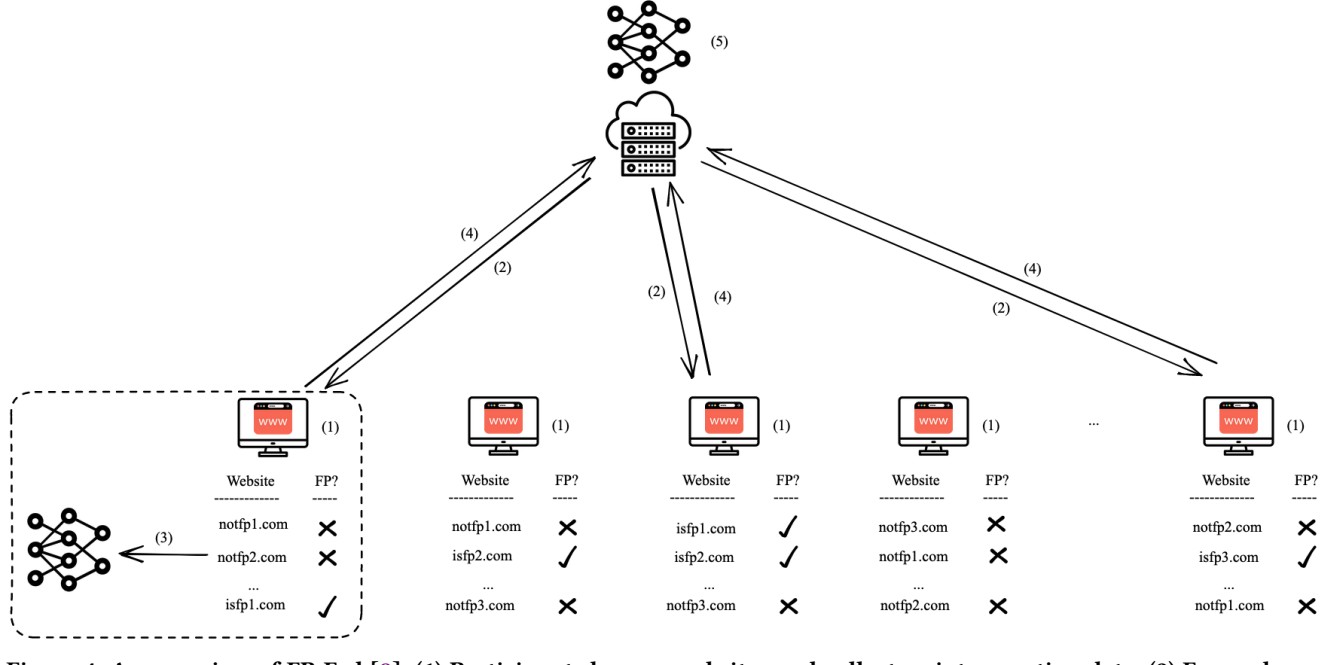

**Figure 4: An overview of FP-Fed [8]: (1) Participants browse websites and collect script execution data. (2) For each round of training, the server sends the previous round's global model parameters to a few selected participants. (3) These selected participants train a local model based on the collected local script execution data, and (4) send the local model updates to the server, (5) which aggregates them and adds differentially private noise to the aggregate to form the global model of the current round. Steps (2) to (5) are then repeated until the model converges.**

    (b) Extracts high-level features for each script loaded on any visited website (e.g., length of string argument, number of values in list returned, etc.);

    (c) Assigns seed ground truth labels (fingerprinting/non-fingerprinting) to each script, according to a high-precision heuristic;

    (d) Participates in a differentially private federated feature pre-processing phase that normalizes extracted high-level features in a privacy-preserving way.

(3) At each round, the server sends the previous round's global model parameters to a subset of participants selected.

(4) Participants initialize a local model with the previous round's global model parameters and train their local model with their locally collected data.

(5) Participants send the model updates to the server.

(6) The server aggregates the model updates, adds statistical noise, and computes the global model parameters for the next round.

(7) Steps 2) to 5) are repeated over multiple rounds until model convergence;

Once the global model converges in FP-Fed, training stops, and all participants can use the trained global model for on-device browser fingerprinting detection by all participants. FP-Fed can be run regularly (e.g., once every few weeks) so that the model can be updated to learn the latest fingerprinting behaviors.

| Type of FP | Real User | Automated |
|---|---|---|
| Canvas | 629 | 177 |
| Canvas Font | 40 | 24 |
| Audio | 215 | 107 |
| WebRTC | 85 | 35 |
| **Total** | **695** | **210** |

**Table 5: Number of fingerprinting scripts captured by real user browsing session vs. automated crawls. *NB:* The same script can perform multiple types of fingerprinting.**

## B Human Intelligence Task (HIT)

In Figure 5, we display the Human Intelligence Task (HIT), i.e., the ad, used to recruit participants from the Amazon mTurk platform. Specifically, prospective participants were provided with some task instructions, an information sheet describing in layman terms the purpose of the data collection and the types of data being collected and a consent form to ensure they have read and understood their data rights and task requirements. They were given a password to authenticate themselves with the Chrome extension used for data collection and were told to enter the "Task Completion Code", which was provided to them by the extension once they completed browsing the assigned websites. For brevity, a significant portion of the consent form has been left out from Figure 5.

## C  Prevalence of Types of Fingerprinting

In Table 5, we report the number of each type of fingerprinting script captured by real user browsing sessions and compare it with the scripts collected from the automated crawl.

Overall, similar to the general prevalence of fingerprinting scripts, we note no major difference between the presence of each type of fingerprinting. Specifically, in both the real user browsing sessions and the automated crawl, Canvas FP is observed to be the most popular form of fingerprinting, followed by Audio, WebRTC, and Canvas Font.

**Task Instructions** (Click to collapse)

We are conducting a study on browser fingerprinting (https://developer.mozilla.org/en-US/docs/Glossary/Fingerprinting), which is an advanced online tracking technique on the web. We want to understand how prevalent this technique is on various popular websites. To that end, we are conducting this user study, which you have volunteered to participate in. Please first read the Participation Information Sheet 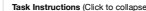 then fill in the Consent Form below. Next, proceed to install the chrome extension from the link provided below. Enter the password you find below into the extension and a file named **sites_to_browse.txt** will be automatically downloaded. Next, open that file and you will see a list of websites. Your task is to visit and interact with each website in that list, by clicking on at least 10 links or buttons you find on each website. Wherever possible, visit login pages (but do not login), accept cookie consents, and solve CAPTCHAs when browsing the websites. After visiting all of the websites, click on the **Done browsing all websites** button and you will receive a task completion code to paste into the box below. Lastly, you can proceed to uninstall the extension and you will receive credit for completing the task. If necessary, please find more detailed instructions here

**Consent Form**

**Please complete this form after you have read the Information Sheet and/or listened to an explanation about the research.**

Title of Study: Collecting Real World Website Data for Browser Fingerprinting Detection

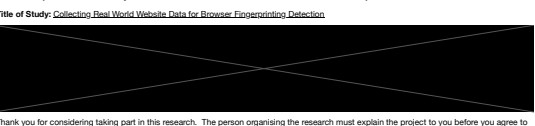

Thank you for considering taking part in this research. The person organising the research must explain the project to you before you agree to take part. If you have any questions arising from the Information Sheet or explanation already given to you, please ask the researcher before you decide whether to join in. You will be given a copy of this Consent Form to keep and refer to at any time.

**I confirm that I understand that by ticking/initialling each box below I am consenting to this element of the study. I understand that it will be assumed that unticked/initialled boxes means that I DO NOT consent to that part of the study. I understand that by not giving consent for any one element that I may be deemed ineligible for the study.**

|  |  | Tick Box |
|---|---|---|
| 1. | I confirm that I have read and understood the Information Sheet for the above study. I have had an opportunity to consider the information and what will be expected of me. I have also had the opportunity to ask questions which have been answered to my satisfaction and would like to take part in the above data collection project. | ☐ |
| 2. | I consent to participate in the study. I understand that my personal information, namely the websites I visit (*restricted to websites in the list provided to me*), and the data from the websites (code executed by websites and browser features accessed by the website *restricted to the list provided*) will be used for the purposes explained to me. I understand that according to data protection legislation, 'performance of a task in the public interest' will be the lawful basis for processing. | ☐ |
| 3. | **Use of the information for this project only**
I understand that all personal information will remain confidential and that all efforts will be made to ensure I cannot be identified.
I understand that my data gathered in this study will be pseudonymized and stored securely. It will not be possible to identify me in any publications. | ☐ |
| 4. | I understand that my information may be subject to review by responsible individuals from ▮▮▮▮▮▮▮ for monitoring and audit purposes. | ☐ |
| 5. | I understand that my participation is voluntary and that I am free to withdraw at any time within 2 weeks of the task deadline as detailed in the MTurk platform without giving a reason.
I understand that if I decide to withdraw, any personal data I have provided up to that point will be deleted unless I agree otherwise.
After 2 weeks, I understand that it will no longer be possible to remove my data as it will have been pseudonymized and the data analysis will already have been conducted. | ☐ |
| 6. | I understand the potential risks of participating and the support that will be available to me should I become distressed during the course of the research. | ☐ |
| 7. | I understand the direct/indirect benefits of participating. | ☐ |
| 8. | I understand that the data will not be made available to any commercial organisations but is solely the responsibility of the researcher(s) undertaking this study. | ☐ |
| 9. | I understand that I will be compensated for the portion of time spent in the study (if applicable) or fully compensated if I choose to withdraw. | ☐ |
| 10. | I understand that the information I have submitted will be published as a research paper but it will not be possible to identify me in the paper. | ☐ |
| 11. | I hereby confirm that I understand the inclusion criteria as detailed in the Information Sheet. | ☐ |
| 12. | I hereby confirm that:
a. I understand the exclusion criteria as detailed in the Information Sheet; and
b. I do not fall under the exclusion criteria. | ☐ |
| 13. | I am aware of who I should contact if I wish to lodge a complaint. | ☐ |
| 14. | I am aware of the use of information for this project and beyond for research purposes.
I would be happy for the data I provide to be archived at a ▮▮▮▮ server for 10 years from the publication of a paper. | ☐ |
| 15. | I voluntarily agree to take part in this study. | ☐ |

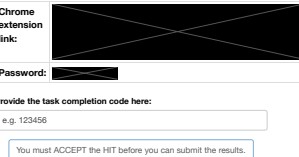

**Chrome extension link:**

**Password:** ▮▮▮▮

**Provide the task completion code here:**

e.g. 123456

You must ACCEPT the HIT before you can submit the results.

