# OpenReview forum: "Beyond the Crawl: Unmasking Browser Fingerprinting in Real User Interactions"
_ACM.org/TheWebConf/2025/Conference — WWW 2025 Poster_

### Official Review · Reviewer_i2Q7 · 2024-11-23

**Novelty:** 5
**Technical Quality:** 6

**Review:**

In this study, researchers conduct a user study that investigates the prevalence of browser fingerprinting in real-world browsing sessions, comparing it to data collected through automated crawls. The study highlights the limitations of automated crawls in accurately capturing fingerprinting behaviors due to their inability to fully replicate human interactions.

Originality and Significance:
- The proposed study addresses a critical gap in fingerprinting research as previous studies primarily relied on automated crawls, which this study demonstrates to be insufficient for capturing the true extent of fingerprinting practices.
- Empirically validates the hypothesis that automated crawls underestimate fingerprinting prevalence. This finding is significant as it underscores the need for more accurate methods of fingerprinting detection.
- The study also introduces the concept of “Miss Percentage” to quantify the extent to which specific website categories are more likely to have undetected fingerprinting. This provides valuable insights for future research and the development of targeted mitigation strategies.
- Explores the potential of federated learning for privacy-preserving fingerprinting detection. This approach offers a promising avenue for training accurate detection models while protecting user privacy.

Quality and Clarity:
- The paper is well-structured and clearly written. It follows logical structure, making it easy to follow the research methodology and findings.
- Provides comprehensive background information on browser fingerprinting, detection techniques, and federated learning. This provides readers with necessary information to follow the rest of the paper.
- Presents detailed methodology, including participant recruitment, data collection, and analysis techniques.
- Uses clear and informative figures and tables to present results effectively.

Pros:
- Novel approach to data collection using a user study with a Chrome extension. This approach captures real-user interaction that automates crawls miss.
- Thorough analysis of the discrepancies between real-user data and automated crawl data.
- Identification of potential new fingerprinting vectors not detected in previous studies. This discovery expands our understanding of the evolving techniques used for browser fingerprinting.
- Demonstrates the efficacy of federated learning for training accuracy and privacy-preserving fingerprinting detection models

Cons:
- Limited scope of website coverage. While the study covers a significant number of websites, it is still a limited sample of the entire web.
- Relatively small participant pool. Although the objective of the study focuses on comparing methodologies rather than drawing population-level conclusions, a larger participant pool could enhance the generalizability of the findings.
- Reliance on a specific heuristic for fingerprinting detection. While the heuristic is established and reliable, future work could explore the use of machine learning classifiers for potentially higher recall.
- Observed large standard deviations in the federated learning evaluation. This suggests the need for further research on larger dataset to confirm the model’s robustness.

**Questions:**

- Regarding the scope of the websites: The study focuses on 3000 top-ranked websites. While this is a significant number, it represents a subset of the websites visited daily by users. Could the authors discuss the potential implications of expanding the study to include a wider range of websites, including those with lower traffic for specific functionalities (e.g., online gaming, financial platforms)? How might the prevalence and types of fingerprinting techniques differ across various website categories and traffic levels?

- Addressing the large standard deviations observed in the federated learning evaluations: The evaluation of FP-Fed on real-user data shows promising results but also exhibits relatively large standard deviations in the AUPRC scores. Could the authors provide further insights into the factors contributing to these variations? How might these deviations be addressed in future research, for example, by exploring different model architectures, aggregating methods, or data processing techniques with the federated learning framework?

- Investigating newly identified fingerprinting vectors: The study intensifies several potential new fingerprinting vectors, particularly related to the audio and WebRTC APIs. Could the authors elaborate on the specific features and attributes of these APIs that make them suitable for fingerprinting? How might these fingerprinting techniques be further investigated and characterized? What are the potential privacy implications of these newly identified vectors, and what mitigation strategies could be considered?

- Clarifying the decision to provide participants with a predefined list of websites: The study design involved providing participants with a list of websites to visit instead of monitoring natural browsing sessions. While this approach addresses privacy concerns, could it potentially introduce bias in the data collection? Could the authors discuss alternative data collection methods that might balance privacy preservation with a more realistic representation of user browsing behavior? For example, could differential privacy techniques be employed to obfuscate individual website visits while still allowing for analysis of fingerprinting prevalence across broader browsing patterns?

- Regarding bot detection scripts: The study finds that bot detection scripts are significant obstacle for automated crawlers. Could the authors elaborate on the types of bot detection mechanisms encountered and their effectiveness in blocking the crawlers? Are there specific strategies or techniques that could be employed to improve the crawler’s ability to circumvent bot detection while still maintaining ethical considerations?

**Ethics Review Description:**

I'm not 100% sure if the ethical considerations are raised but based on my understanding, the authors claim that only pre-processed features are used and not the raw data. Also, while federated learning protects raw telemetry from disclosure, they still required access to this data for their experiments. If I'm missing something here, please feel free to let me know, I can update this accoridngly.

**Ethics Review Flag:**

Yes

**Reviewer Confidence:**

3: The reviewer is confident but not certain that the evaluation is correct

**Scope:**

4: The work is relevant to the Web and to the track, and is of broad interest to the community

---

### Official Review · Reviewer_AcHi · 2024-11-29

**Novelty:** 5
**Technical Quality:** 5

**Review:**

### Summary

This paper presents an analysis of the fingerprinting websites from real-user browsing sessions. It conducts a user study across 3000 top-ranked websites over a 10-week period, collecting real-user browsing sessions in contrast to the automated crawled dataset. Their analysis shows that around 47.8% of fingerprinting websites identified by their real-world user-browsed dataset are missed by the automated crawling dataset due to the requirement of user interactions, including page authentication, bot detection, and cookie banners. Also, they discovered potential new fingerprinting vectors from their real-user browsing sessions that haven't been found by automated crawls. They trained machine learning models deploying FP-fed. As an improvement, they pre-trained the local models on public data and then fine-tuned them on private data to enhance the performance. Their models can preserve privacy and achieve comparable or better performance than non-private models trained on automated crawled datasets.

### Pros

1. Quality:
   This paper has a good overall quality. It is generally easy to follow. The study it conducted is quite interesting, and it indeed reveals an issue for existing research in fingerprinting regarding automated crawls and shows a solution with their user study.

2. Clarity

   The paper is generally clear and easy to follow.

3. Originality

   The paper has an impressive observation on the defects of automated crawl datasets in the fingerprinting issue, which motivated them to collect and analyze real-user data, therefore showing good novelty.

4. Significance
   This paper has a sound motivation, and their study does make sense and shows good significance. It observed that existing studies on fingerprinting issues that rely on automated crawled data have limitations because some sessions require human interaction, which might make the automated crawlers miss these data. This insight is quite interesting. They conducted a user study collecting and analyzing real-user-browsed sessions and found that almost half of the fingerprinting websites identified by user browsing are missed by automated crawls, and they also discovered potential new fingerprinting vectors.

### Cons

1. Implementation bug of Chrome extension. As the authors acknowledged in section 4.1, their Chrome extension includes bugs. It would be better if they could conduct experiments with the bug fixed or provide a more in-depth analysis of the bug.

2. Uncleared discussion on the number of fingerprinting websites missed by automated crawls and identified from user browsing in section 4.1. In this section, the paper mentioned that there are 225 websites missed by automated crawls. It seems to be implied in lines from 546 to 547 that it is summed up by 108 and 117. However, as mentioned in lines 560 and 561,  it seems these 117 websites are detected by automated crawlers. It looks like the discussion here is a bit confusing, and the relationship between these numbers should be clarified.

3. Regarding the websites of interest, the paper excludes websites with categories such as adult themes, gambling, etc. However, simply excluding those websites may lead to bias in the results, as these websites are usually suspicious and may be prevalent in fingerprinting.

4. Detailed evaluation results should be presented. Though AUPRC is a good metric, because the paper discussed many detailed advantages of proposed models, it would be better to also attach the associated metrics directly. For example, the paper mentioned several times minimizing false positives in their model. While it somehow could be reflected by AUPRC, it's better to present the results and analyze them explicitly. Also, the paper mentioned at the end of section 5.2 that the standard deviations are large, but no in-depth analysis or data are presented.

**Questions:**

1. Could you please clarify the 225 websites missed by the automated crawler? It seems the discussion in section 4.1 implies that it is from 108 + 117. However, in lines 560 and 561, it seems these 117 websites are detected by automated crawlers. Can you explain the relationship of these numbers?

2. Regarding the model training, does the pretraining with public data improve a lot? What if not pre-trained with public data? Also, what would the result be if a centralized model is directly trained on the real-user data?

3. The paper only presents AUPRC results in Table 4. Could you present other important metrics like false positives that were not included in the evaluation results or explain their absence, especially given the paper's emphasis on minimizing false positives?

**Reviewer Confidence:**

3: The reviewer is confident but not certain that the evaluation is correct

**Scope:**

4: The work is relevant to the Web and to the track, and is of broad interest to the community

---

### Official Review · Reviewer_9ttL · 2024-12-01

**Novelty:** 5
**Technical Quality:** 4

**Review:**

*Summary*

This paper investigates browser fingerprinting by analyzing data from real user browsing sessions, revealing significant discrepancies between automated web crawlers and real user interactions. It highlights the limitations of crawlers in detecting fingerprinting websites and explores the potential of federated learning to improve detection using real user data.

*Strengths*

1. The paper effectively demonstrates the performance gap in script collection between automated crawlers and real user interactions.

2. The findings show that real user sessions capture a higher number of fingerprinting websites compared to automated crawlers, emphasizing the value of real-world data.

*Weaknesses*

1.The main findings could be expressed more clearly to strengthen the impact of the study.

2.The paper lacks an in-depth analysis of the root causes behind the discrepancies between automated crawlers and real users.

*Detailed Comments*

The observation that real user interactions capture significantly more fingerprinting scripts than automated crawlers is intriguing. However, the explanation attributing this to bot detection mechanisms feels incomplete. It would have been valuable to experiment with techniques to bypass bot detection during automated crawling to better support this claim.

In Section 3.2, it is stated that each participant visited 506 unique URLs, which is five times the number of URLs visited by automated crawlers in prior studies. However, this statement is unclear. Upon reviewing the referenced paper [31], it seems that a single crawler visited approximately 100 unique URLs. It is unclear whether comparing the number of URLs visited by multiple participants (30) to a single crawler is meaningful. To improve clarity, a comparison using more standardized metrics or presenting the data in a table with precise numbers would be more appropriate.

In Section 4.4, the abstract claims the discovery of potential new fingerprinting vectors, but the AudioContext API and WebRTC API are already discussed in [27]. Furthermore, according to [31], JavaScript APIs with a call ratio exceeding 15 are highlighted, while the Navigator API, proposed in this paper, shows a much lower ratio. The paper needs to provide a clearer explanation of what constitutes a "new fingerprinting vector" and substantiate its claims.

Regarding Section 5, the comparison between FP-Fed on Real-User Sessions and centralized models is interesting. However, it appears that the federated model should focus on fine-tuning a pre-trained model using local datasets. Beyond model performance, the benefits of this approach compared to existing FP-Fed should be discussed more explicitly.

*Minor Comments*

1. For statements referencing percentages or ratios (e.g., "almost half," "2.3X more"), it would be helpful to include exact figures alongside relative terms.

2. Consider presenting numerical comparisons in tables or figures to enhance clarity and accessibility.

**Questions:**

1. In Section 3.2, how do you justify comparing the number of unique URLs visited by multiple participants (30) to those visited by a single automated crawler? Would presenting standardized metrics or aggregated comparisons improve clarity?

2. What specifically qualifies the Navigator API as a "new fingerprinting vector," given that its call ratio is significantly lower than the threshold highlighted in [31]?

3. Beyond improved model performance, what additional benefits does the federated learning approach on real user data provide compared to centralized models?

4. How do the findings on AudioContext API and WebRTC API as fingerprinting vectors differ from prior work ([27]), and how do these validate the claim of discovering new vectors?

**Reviewer Confidence:**

4: The reviewer is certain that the evaluation is correct and very familiar with the relevant literature

**Scope:**

4: The work is relevant to the Web and to the track, and is of broad interest to the community

---

### Official Review · Reviewer_3SnG · 2024-12-02

**Novelty:** 3
**Technical Quality:** 2

**Review:**

This paper presents an interesting study that compares the prevalence of browser fingerprinting in real user browsing sessions with automated crawls. The authors have taken appropriate steps to ensure the privacy of their participants. The paper is also well-written and easy to follow.

## Pros
- The study is the first of its kind to investigate browser fingerprinting in real user interactions.
- The authors have taken appropriate steps to ensure the privacy of their participants.
- The paper is well-written and easy to follow.

## Cons
- Methodology lacks details.
- Comparison of real-user data and the automated crawler is not convincing.
- Study is only focused on Google Chrome.

The findings of the study highlight the limitations of automated crawlers for detecting browser fingerprinting. They also demonstrate the feasibility of using federated learning to train privacy-preserving fingerprinting detection models.

However, while the research question is relevant and timely, the paper suffers from several methodological limitations that weaken its conclusions and claims of scalability and comprehensiveness. Overall, I will support this paper for major revision if the authors could address the following concerns and questions:

---

## Major Concerns

### 1. Scale and Generalizability
The authors claim their work paves the way for "large-scale deployment" of fingerprinting detection. However, this claim is not supported by the study's design. A user study with only 30 participants with pre-determined websites from a pool of 5k total websites is not considered neither scalable nor large-scale.  Furthermore, limiting the study to a single browser (Google Chrome) further restricts the generalizability of the findings. Prior research (e.g. Browserprint paper [4]), has demonstrated significant differences in fingerprinting patterns across browsers. Authors either need to change their scalability and large-scale claim or, the study should include multiple browsers and improve the way target urls are being picked and analyzed.

---

### 2. Limitations and Bias in Crawler Script
The comparison between real-user browsing and the automated crawl is flawed. The crawler only visits the main page of each website and doesn't explore sub-pages, resulting in a biased comparison. The authors could have provided the crawler with the exact list of URLs visited by the real users to ensure a fair comparison. Additionally, the paper lacks detailed information about the crawler script used. While it is mentioned in Section 4.1, crucial details about its features are missing, and the script itself is not available for scrutiny. This lack of transparency makes it difficult to assess the validity of the comparison and raises concerns about potential bias. To support your paper’s claim, the real-user results should be compared to either prior work tools, or a more detailed and publicly available crawler script.

---

### 3. Restricting User Browsing
Instead of capturing natural browsing behavior, the authors provided participants with a list of websites and instructed them to visit 10 sub-pages. This approach raises another concerns about scalability and bias. The authors argue that this decision was necessary to protect user privacy, but anonymizing collected data could have achieved the same objective without sacrificing ecological validity.

---

## Recommendations
- **Improve the automated crawl:** Provide a more detailed explanation of the crawler script, including its features and limitations. Make the script publicly available for transparency and reproducibility. Ensure a fair comparison by having the crawler visit the same URLs as the real users.
- **Address the limitations of scale and generalizability:** Include multiple browsers to support the claims of large-scale deployment and improve generalizability.
- **Re-evaluate the data collection methodology:** Explore alternative approaches to protect user privacy without restricting natural browsing behavior. Consider anonymizing collected data to reduce concerns about user privacy.

**Questions:**

- What are the "fingerprinting websites" mentioned in the paper? Does a regular website that also has a fingerprinting script is classified as a fingerprinting website?

- What does the crawler script do when it faces a cookie consent form? Older prior work might not have taken this into their results but accepting/rejecting that consent form is not complicated and will highly likely impact the results.

- The authors mention that they did not verify if participants visited at least ten subpages on each website. Could this have impacted the results of the study? If so, how?

**Reviewer Confidence:**

4: The reviewer is certain that the evaluation is correct and very familiar with the relevant literature

**Scope:**

4: The work is relevant to the Web and to the track, and is of broad interest to the community

---

### Official Review · Reviewer_ERLG · 2024-12-03

**Novelty:** 5
**Technical Quality:** 4

**Review:**

The paper compares browser fingerprinting in automated crawls to when real users interact with websites. The paper discusses what aspects of user interactions lead to these differences, the use of different APIs, and the website categories where the automated crawler missed the most websites that did fingerprinting. The paper also evaluates a modified existing tool (FP-Fed) on the data they collected from real user interactions with sites.


Strengths of the paper
* Provides an evaluation of FP-Fed (with additional pre training) on data collected by real users
* Shows that FP-Fed trained on real user browsing with high privacy settings can provide comparable performance to a centralized model trained only on automated crawl data
* Highlights the limitations of exclusively using automated crawls in the research of browser fingerprinting
* Does a good job discussing ethics and the steps taken to protect and compensate the human subjects

Weaknesses of the paper
* Due to an issue with the extension used for the data collection from real users, it is unclear to me how representative the percentages and counts in the paper are. The implications of this bug on the findings of the paper could be more fully discussed.
* The paper left me with some unanswered questions about how comparable the real user browsing and automated crawl were

**Questions:**

* Why did the automated crawl not visit the same subpages visited by the real users for the automated crawl? Were the comparisons in 4.1 based on just the pages of websites that were visited by both the real users and the automated crawl?
* What parts of the paper/findings were impacted by the bug in the real user browsing data collection/how representative are the numbers and percentages reported in the paper?
* What steps were taken to make the automated crawler avoid bot detection/make the crawler as close to a real user beyond just taking actions like scrolling?
* What geographic locations were people recruited from on MTurk? What was the general vantage point of the automated crawler?

**Reviewer Confidence:**

3: The reviewer is confident but not certain that the evaluation is correct

**Scope:**

4: The work is relevant to the Web and to the track, and is of broad interest to the community